# Boosting the Photocatalysis of Plasmonic Au-Cu Nanocatalyst by AuCu-TiO_2_ Interface Derived from O_2_ Plasma Treatment

**DOI:** 10.3390/ijms241310487

**Published:** 2023-06-22

**Authors:** Bin Zhu, Xue Li, Yecheng Li, Jinglin Liu, Xiaomin Zhang

**Affiliations:** 1College of Environmental Sciences and Engineering, Dalian Maritime University, Dalian 116024, China; binzhu@dlmu.edu.cn (B.Z.);; 2Laboratory of Plasma Physical Chemistry, Dalian University of Technology, Dalian 116024, China; 3Dalian National Laboratory for Clean Energy, Dalian Institute of Chemical Physics, Chinese Academy of Sciences, Dalian 116023, China

**Keywords:** plasma treatment, Au catalyst, plasmonic catalyst, CO oxidation

## Abstract

Plasmonic gold (Au) and Au-based nanocatalysts have received significant attention over the past few decades due to their unique visible light (VL) photocatalytic features for a wide variety of chemical reactions in the fields of environmental protection. However, improving their VL photocatalytic activity via a rational design is prevalently regarded as a grand challenge. Herein we boosted the VL photocatalysis of the TiO_2_-supported Au-Cu nanocatalyst by applying O_2_ plasma to treat this bimetallic plasmonic nanocatalyst. We found that O_2_ plasma treatment led to a strong interaction between the Au and Cu species compared with conventional calcination treatment. This interaction controlled the size of plasmonic metallic nanoparticles and also contributed to the construction of AuCu-TiO_2_ interfacial sites by forming AuCu alloy nanoparticles, which, thus, enabled the plasmonic Au-Cu nanocatalyst to reduce the Schottky barrier height and create numbers of highly active interfacial sites. The catalyst’s characterizations and density functional theory (DFT) calculations demonstrated that boosted VL photocatalytic activity over O_2_ plasma treated Au-Cu/TiO_2_ nanocatalyst arose from the favorable transfer of hot electrons and a low barrier for the reaction between CO and O with the construction of large numbers of AuCu-TiO_2_ interfacial sites. This work provides an efficient approach for the rational design and development of highly active plasmonic Au and Au-based nanocatalysts and deepens our understanding of their role in VL photocatalytic reactions.

## 1. Introduction

The plasmonic nanostructures of metals (mainly copper, silver, and gold) show significant promise in the conversion of solar energy toward chemical energy [1,2]. Under visible light (VL) illumination, plasmonic metallic nanostructures can strongly interact with resonant photons via localized surface plasmon resonance (LSPR), thus effectively driving a wide variety of chemical reactions. For example, the application of VL photocatalysis in CO oxidation has attracted a great deal of attention because it allows for the efficient oxidation of CO into CO_2_ at ambient temperatures and solar light irradiation, showing great potential in air purification and the selective oxidation of CO in excess hydrogen. Among the plasmonic metals, gold (Au) is widely used to construct supported plasmonic Au nanocatalysts and draws increasing attention in CO oxidation, mainly due to its high VL absorption cross-sections as well as superior inherent catalytic activity [3,4,5].

Improving VL photocatalytic activity via a rational design is prevalently regarded as a grand challenge for the development of supported plasmonic Au nanocatalysts. Great efforts have been devoted to enhancing the VL photocatalytic activity by modulating the features (such as Au nanoparticle size, support nature, and Au valence state) of Au nanocatalysts [6,7,8]. An alternative avenue is to design plasmonic Au-based nanocatalysts, and previous studies have reported that TiO_2_-supported Au-Ag and Au-Cu bimetallic nanocatalysts could exhibit a much higher VL photocatalytic activity compared to the monometallic Au/TiO_2_ nanocatalysts [9,10]. This enhanced VL photocatalytic activity predominately lies in the fact that bimetallic plasmonic catalysts provide highly active metal–support interfaces for photocatalytic reactions [9,11]. Compared with monometallic Au nanocatalysts, the introduction of a secondary metal makes the metal–support interfaces of Au-based bimetallic samples possess a lower Schottky barrier height due to the formation of alloy metals, facilitating the interfacial transfer of hot electrons in photocatalytic reactions [9,12]. Moreover, the electronic and structural features of metal–support interfaces could also be modulated with the presence of a secondary metal in Au-based bimetallic nanocatalysts [13,14], which can improve the adsorption and activation behaviors of reactants at interfaces and, thus, boost photocatalytic reactions.

To construct highly active metal–support interfaces, plasmonic Au-based bimetallic nanocatalysts require efficient treatment/activation [9,13,15]. Calcination (commonly ≥200 °C) has been intensively reported to treat the supported plasmonic Au-based nanocatalysts [16,17] because their thermal treatment could reduce the Au into plasmonic Au nanoparticles and also exhibit great advantages in creating active Au-support interfacial sites by tuning the metal–support interactions [18,19,20]. However, Au nanoparticles tended to coalesce over the support surface during the calcination because of the relatively low Tammann temperature of Au nanoparticles [21,22]. The agglomeration of Au nanoparticles undermined the number of interfacial sites while also causing the separation of Au and the secondary metal, thus suppressing the formation of highly active metal–support interfaces.

Cold plasma is a promising approach when tackling the problems encountered by thermal treatment [23,24], owing to its high reactivity at low temperatures [25]. It has been reported that cold plasma treatment enables Au nanocatalysts to obtain a controlled Au nanoparticle size [15,26] because large numbers of reactive species in cold plasma (energetic electrons, radicals, etc.) effectively reduce cationic Au species at low temperatures (even room temperature). The dispersed metallic nanoparticles on the support, which play the role of expanding the plasma region over the nanocatalysts surface [27,28,29], enable efficient interaction between the reactive species and the surface of Au-based nanocatalysts. This could establish unique chemical environments over the nanocatalyst’s surface, therefore endowing cold plasma with a strong ability to restructure surface species and tailoring the features of metal–support interfaces during the treatment [30].

Herein, we propose a strategy to create highly active interfacial sites for the VL photocatalytic reaction by exploiting O_2_ plasma to treat the plasmonic Au-Cu/TiO_2_ nanocatalyst. The VL photocatalytic oxidation (PCO) of CO was used to evaluate the performance of plasmonic nanocatalysts due to its high activity at mild conditions and was widely applied in the fields of environmental protection (for example, indoor air purification) and energy conversion (for example, CO removal from H_2_-rich gas). It was found that the Au-Cu/TiO_2_ sample treated by O_2_ plasma outperformed in VL photocatalytic the monometallic nanocatalysts (Au/TiO_2_ or Cu/TiO_2_) or conventional calcined Au-Cu/TiO_2_ nanocatalyst. We demonstrated how O_2_ plasma treatment obtained a strong interaction between the Au and Cu species, which played a pivotal role in controlling the size of plasmonic nanoparticles, producing AuCu alloy nanoparticles, and, thus, constructing AuCu-TiO_2_ interfacial sites. The catalyst characterizations and density functional theory (DFT) calculations suggested that O_2_ plasma enabled the Au-Cu/TiO_2_ sample to boost the transfer of hot electrons and a lower barrier for the reaction between CO and dissociated O_2_ by reducing the Schottky barrier height and creating numbers of highly active AuCu-TiO_2_ interfacial sites.

## 2. Results

### 2.1. Studies of VL PCO of CO

The plasmonic Au-Cu/TiO_2_ bimetallic nanocatalyst, as well as the monometallic Au and Cu nanocatalysts, were applied to the VL PCO of CO. The Au-Cu/TiO_2_ nanocatalyst (loadings of Au and Cu are 1 wt.% and 0.5 wt.%, respectively) treated by O_2_ plasma and calcination (200 °C) were denoted as (Au_1_Cu_0.5_/TiO_2_)_OP_ and (Au_1_Cu_0.5_/TiO_2_)_C200_, respectively. For comparison, the Au/TiO_2_ (Au loading of 1 wt.%) and Cu/TiO_2_ (Cu loading of 1 wt.%) nanocatalysts treated by O_2_ plasma were prepared and denoted as (Au_1_/TiO_2_)_OP_ and (Cu_1_/TiO_2_)_OP_, respectively. Figure 1a shows the VL photocatalytic activity of plasmonic nanocatalysts. The (Au_1_/TiO_2_)_OP_ obtained a CO conversion (*X*_CO_) of 74%, whereas the (Cu_1_/TiO_2_)_OP_ showed no photocatalytic activity for CO oxidation due to its weak LSPR effect and poor intrinsic catalytic activity at room temperature. Similarly, in the absence of Au, TiO_2_ treated by O_2_ plasma showed no activity for CO oxidation under VL irradiation. By contrast, (Au_1_Cu_0.5_/TiO_2_)_OP_ achieved the highest *X*_CO_ (86%) among the plasmonic samples under VL irradiation, indicating a strong synergistic effect between Au and Cu. In addition, the activity of (Au_1_/TiO_2_)_OP_ and (Au_1_Cu_0.5_/TiO_2_)_OP_ both exhibited a decline without VL irradiation (see Appendix A). It could be noted that the Au/Cu ratio showed an obvious influence on the performance of plasmonic Au-Cu/TiO_2_ nanocatalysts. We investigated the effect of the Au/Cu ratio on photocatalytic activity by keeping Au loading at 1 wt.%. With the decrease in Au/Cu loading, the mass-specific reaction rates of the samples treated by O_2_ plasma appeared as a peak curve variation (see Appendix A), and a maximum value of 5.3 mol g^−1^ h^−1^ was achieved by the (Au_1_Cu_0.5_/TiO_2_)_OP_. This indicates that the Au/Cu ratio of two is an optimal ratio for the plasmonic Au-Cu/TiO_2_ nanocatalysts. The synergism between Au and Cu highly depends on the treatment approaches. As shown in Figure 1a, the (Au_1_Cu_0.5_/TiO_2_)_C200_, which was treated by conventional calcination, even presented lower photocatalytic activity (*X*_CO_ of 58%) than the (Au_1_/TiO_2_)_OP_. Further, for the (Au_1_Cu_0.5_/TiO_2_)_C200_, an obvious induction period can be observed in Figure 1a. The (Au_1_Cu_0.5_/TiO_2_)_OP_ exhibited a much shorter induction period than the (Au_1_Cu_0.5_/TiO_2_)_C200_, as the strong interaction between Au and Cu facilitated auto-reduction during the VL PCO of CO [9,13]. Apparently, for the plasmonic Au-Cu/TiO_2_ nanocatalysts, O_2_ plasma treatment enabled the synergism between Au and Cu and, thus, led to a boost in photocatalytic activity, which suggested the probability of cost reductions in Au-Cu/TiO_2_ nanocatalysts by partially replacing Au with Cu. To confirm this assumption, we further compared the performance of (Au_1_/TiO_2_)_OP_ and (Au_0.5_Cu_0.25_/TiO_2_)OP in the VL PCO of CO (see Figure 1b). During 360 min of continuous test, the (Au_0.5_Cu_0.25_/TiO_2_)_OP_ exhibited comparable activity and stability to the (Au_1_/TiO_2_)_OP_, although it only possessed about half the Au loading of the (Au_1_/TiO_2_)_OP_. The boosted photocatalytic performance of (Au_1_Cu_0.5_/TiO_2_)_OP_ could be attributed to the O_2_ plasma treatment, which enabled a strong interaction between the Au and Cu species and, thus, the construction of unique metal–support interface structures. Hence, the electronic and structural states of the plasmonic nanocatalysts were investigated and compared.

### 2.2. Electronic State of the Plasmonic Nanocatalysts

The surface electronic state of plasmonic nanocatalysts was measured by XPS. In Figure 2, the binding energies of Au 4*f* and Cu 3*d* are deconvoluted, and the results are summarized in Table 1. The Au 4*f* spectra of Au_1_Cu_0.5_/TiO_2_ nanocatalysts were deconvoluted into Au^0^ and Au^δ+^, while the Cu 2*p* spectra contained Cu^2+^, Cu^+^, and Cu^0^ peaks. Further, the Cu^0^/Cu^+^ ratio was evaluated by analyzing the Cu LMM spectra. As listed in Table 1, the contents of Au^δ+^ and Cu^2+^ for the fresh (Au_1_Cu_0.5_/TiO_2_)_OP_ were 43% and 35%, respectively. Due to the auto-reduction in cationic Au and Cu species in the induction period, the contents of Au^δ+^ and Cu^2+^ for the used (Au_1_Cu_0.5_/TiO_2_)_OP_ decreased to 29% and 32%, respectively. The same phenomenon was also observed for (Au_1_Cu_0.5_/TiO_2_)_C200_. Interestingly, compared with (Au_1_Cu_0.5_/TiO_2_)_C200_, the (Au_1_Cu_0.5_/TiO_2_)_OP_ possessed a higher Au^δ+^ content but lower Cu^2+^ content. This probably originated from the intimately interacted Au and Cu species in the (Au_1_Cu_0.5_/TiO_2_)_OP_, as their strong interaction enabled the Cu species to obtain a higher electron density than the Au species [31]. Without the interaction between the Au and Cu species, the (Au_1_/TiO_2_)_OP_ and (Cu_1_/TiO_2_)_OP_, respectively, exhibited lower Au^δ+^ and higher Cu^2+^ contents than the (Au_1_Cu_0.5_/TiO_2_)_OP_ (see Appendix A). Further, compared with (Au_1_/TiO_2_)_OP_ and (Cu_1_/TiO_2_)_OP_, the binding energies of Au 4*f* and Cu 2*p* of the (Au_1_Cu_0.5_/TiO_2_)_OP_ exhibited blue- and red-shifts (see Table 1 and Appendix A), respectively. These shifts were induced by the interaction between Au and Cu species, which led to a shift in the electron density from the Au to Cu species. Table 1 also shows that O_2_ plasma treatment endowed the samples with a higher Cu^0^/Cu^+^ ratio than the calcined samples, indicating the favorable formation of metallic Cu in the O_2_ plasma-treated sample. The above results suggest that the Cu species in the (Au_1_Cu_0.5_/TiO_2_)_OP_ tended to achieve low valence, which facilitated the creation of AuCu alloy nanoparticles and, thus, AuCu-TiO_2_ interfacial sites. Furthermore, the binding energies of Au 4*f* and Cu 2*p* for the (Au_1_Cu_0.5_/TiO_2_)_OP_ both presented a blue shift compared with (Au_1_Cu_0.5_/TiO_2_)_C200_, indicating the electron deficiency of O_2_ plasma-treated samples. This could be attributed to the fact that the discharge in oxidative O_2_ enabled strong interactions between the Au, Cu species, and TiO_2_ support [15,32]. Considering the generation of large numbers of oxygen species (e.g., O^2+^, O^−^) in O_2_ discharge, O_2_ plasma treatment led to a high content of surface oxygen (O_S_) (see Appendix A). The content of O_S_ decreased after the VL PCO of CO due to the partial consumption of reactive oxygen species during the reaction. Further, we noted that the presence of Cu resulted in higher O_S_ content (see Appendix A), which could probably be attributed to the construction of AuCu-TiO_2_ interfacial sites, which benefited the adsorption and activation of oxygen.

To further investigate the interaction between Au and Cu species in the Au-Cu/TiO_2_ nanocatalysts, the H_2_-TPR of the fresh plasmonic nanocatalysts was performed. Figure 3 shows that (Au_1_/TiO_2_)_OP_ and (Cu_1_/TiO_2_)_OP_ both presented two reduction peaks with increasing temperature from −50 °C to 250 °C. The H_2_ consumption peaks located at low temperatures (35 °C and 32 °C, respectively) could be attributed to a reduction in the surface oxygen species [33], while the peaks located at 65 °C and 123 °C were ascribed to the reduction of Au^δ+^ and Cu^δ+^ species, respectively. Figure 3 also shows that (Au_1_Cu_0.5_/TiO_2_)_OP_ exhibited two obvious reduction peaks at 34 °C and 63 °C, respectively, although it possessed the Au^δ+^ and Cu^δ+^ species. Apparently, the Au^δ+^ and Cu^δ+^ species in (Au_1_Cu_0.5_/TiO_2_)_OP_ only presented one reduction peak at 63 °C, which might be caused by their strong interaction. By contrast, the Au^δ+^ and Cu^δ+^ species in (Au_1_Cu_0.5_/TiO_2_)_C200_ showed a reduction peak at a higher temperature of 87 °C compared to (Au_1_Cu_0.5_/TiO_2_)_OP_, which might originate from its low content of Au^δ+^ and the weak interaction between the Au^δ+^ and Cu^δ+^ species. Further, (Au_1_Cu_0.5_/TiO_2_)_OP_ possessed the largest peak area of surface oxygen reduction among all the samples (see Figure 3), as the strong interaction between Au and Cu species facilitated the adsorption and activation of oxygen by creating large numbers of AuCu-TiO_2_ interfacial sites.

CO adsorption behaviors over the plasmonic nanocatalysts are displayed in Figure 4. The (Au_1_/TiO_2_)_OP_ showed a band at 2107 cm^−1^ due to the adsorption of CO onto the metallic Au [34,35], and (Cu_1_/TiO_2_)_OP_ appeared with the band at 2109 cm^−1^, which could be ascribed to CO adsorption on Cu+ species, due to the weak adsorption ability of the metallic Cu and Cu^2+^ species for CO [36,37]. Furthermore, the (Au_1_/TiO_2_)_OP_ exhibited a much weaker band intensity compared with the (Cu_1_/TiO_2_)_OP_, indicating that the Cu^+^ species possessed a stronger adsorption capacity for CO molecules than metallic Au. The (Au_1_Cu_0.5_/TiO_2_)_C200_ and (Au_1_Cu_0.5_/TiO_2_)_OP_ presented bands at 2106 cm^−1^ and 2112 cm^−1^, respectively, and the adsorption bands for these two samples exhibited a comparable intensity to the monometallic (Cu_1_/TiO_2_)_OP_ (see Figure 4). This suggests that the Cu^+^ species, rather than metallic Au, mainly contributed to CO adsorption in the Au_1_Cu_0.5_/TiO_2_ nanocatalysts. Compared to the (Cu_1_/TiO_2_)_OP_ and (Au_1_Cu_0.5_/TiO_2_)_C200_, the (Au_1_Cu_0.5_/TiO_2_)_OP_ displayed a stronger band intensity, as the strong interaction between the Au and Cu species induced by O_2_ plasma treatment endowed (Au_1_Cu_0.5_/TiO_2_)_OP_ with large numbers of coordinatively unsaturated sites for CO adsorption [15,38]. Figure 4 also shows that the adsorption band of the (Au_1_Cu_0.5_/TiO_2_)_OP_ had a blue shift of 6 cm^−1^ compared with that of the (Au_1_Cu_0.5_/TiO_2_)_C200_. This shift reflects an electron transfer from metallic Au toward Cu^+^ species, owing to the strong interaction between the Au and Cu species (see H_2_-TPR analysis). The CO adsorption behaviors of (Au_1_Cu_0.5_/TiO_2_)_OP_ further confirm the pivotal role of O_2_ plasma treatment in generating a strong interaction between Au and Cu species and constructing highly active AuCu-TiO_2_ interfacial sites.

### 2.3. Structural States of the Plasmonic Nanocatalysts

The morphologies of the plasmonic nanoparticles in different nanocatalysts were observed by HR-TEM. As shown in Figure 5, the metallic nanoparticles on the TiO_2_ surface for the (Au_1_Cu_0.5_/TiO_2_)_OP_, (Au_1_Cu_0.5_/TiO_2_)_C200,_ and (Au_1_/TiO_2_)_OP_ could be clearly observed. The average size of the metallic nanoparticles for the samples obeyed an order of (Au_1_Cu_0.5_/TiO_2_)_OP_ < (Au_1_Cu_0.5_/TiO_2_)_C200_ < (Au_1_/TiO_2_)_OP_, and the (Au_1_Cu_0.5_/TiO_2_)_OP_ achieved the smallest average size of 2.4 nm (see Appendix A). It can be noted that (Cu_1_/TiO_2_)_OP_ possessed the nanoparticles without clear lattice fringes (see Figure 5d). These nanoparticles were probably ascribed to CuO_X_, as the Cu^2+^ species are difficult to be reduced into Cu nanoparticles in the VL PCO of CO. The average size of CuO_X_ nanoparticles in the (Cu_1_/TiO_2_)_OP_ was not counted because of the relatively low contrast between CuO_X_ and TiO_2_ (see Appendix A). The HR-TEM images confirm that the plasmonic nanocatalysts treated by O_2_ plasma could obtain metallic plasmonic nanoparticles with small particle sizes. The interaction between the Au and Cu species also contributed to the controlled size of metallic nanoparticles, as demonstrated by the smaller size of plasmonic nanoparticles in (Au_1_Cu_0.5_/TiO_2_)_OP_ compared to that of (Au_1_/TiO_2_)_OP_.

The HAADF-STEM-EDS technique was used to identify the composition of the metallic plasmonic nanoparticles in (Au_1_Cu_0.5_/TiO_2_)_OP_. The EDS mapping in Figure 6 shows the homogeneous distribution of the Au and Cu elements in a typical nanoparticle, indicating the formation of AuCu alloy nanoparticles in the used (Au_1_Cu_0.5_/TiO_2_)_OP_. Further, the EDS line profiles of the used (Au_1_Cu_0.5_/TiO_2_)_OP_ illustrate that the peaks of the Au and Cu elements almost exhibited the same variation trend (see Appendix A), demonstrating that the AuCu alloy nanoparticles possessed a uniform structure rather than separated structures (core–shell, Janus, etc.). The formation of AuCu alloy nanoparticles in the used (Au_1_Cu_0.5_/TiO_2_)_OP_ could be attributed to the strong interaction between the Au and Cu species, as the hot electrons produced by metallic Au could reduce adjacent cationic Cu into metallic Cu under VL irradiation. The (Au_1_Cu_0.5_/TiO_2_)_C200_ with a weak interaction between the Au and Cu species could only generate minor numbers of AuCu alloy nanoparticles, and thus, the AuCu alloy nanoparticles were difficult to identify from the used (Au_1_Cu_0.5_/TiO_2_)_C200_ in the HAADF-STEM-EDS observation. This suggests that intimately interacting Au and Cu species are essential for obtaining the AuCu alloy nanoparticles.

The above results indicate that the (Au_1_Cu_0.5_/TiO_2_)_OP,_ possessing numerous small AuCu alloy nanoparticles, could create larger numbers of interfacial sites between metallic nanoparticles and the TiO_2_ support among all the samples. To confirm this point, we estimated and compared the number of interfacial sites that were located at the metal–support interface based on the morphology and average size of plasmonic nanoparticles. A combination of the DFT-based Wulff Construction and physical model reconstruction was used to calculate the proportion of the perimeter metal atoms (see Appendix A) [30]. Table 2 lists that (Au_1_Cu_0.5_/TiO_2_)_OP_ possessed a higher proportion of perimeter metal atoms (6.0%) compared with the (Au_1_/TiO_2_)_OP_ (5.3%) and (Au_1_Cu_0.5_/TiO_2_)_C200_ (4.7%), demonstrating that a larger number of interfacial sites were constructed in the (Au_1_Cu_0.5_/TiO_2_)_OP_.

### 2.4. Photo-Response Properties of the Plasmonic Nanocatalysts

The LSPR effect of plamonic nanocatalysts was measured by UV-vis DRS. Figure 7a shows that a strong light absorption band at ca. 568 nm could be clearly observed for the used (Au_1_/TiO_2_)_OP_ due to the LSPR of metallic Au nanoparticles [15]. The fresh (Au_1_/TiO_2_)_OP_ only presented a weak light absorption band because of its low Au^0^ content after O_2_ plasma treatment, which was consistent with the XPS results. The fresh and used (Cu_1_/TiO_2_)_OP_ both appeared as weak absorption bands after 600 nm, indicating the weak light absorption ability of (Cu_1_/TiO_2_)_OP_, owing to their low Cu^0^ content and relatively low absorption cross-section of Cu nanoparticles [39,40]. The (Au_1_Cu_0.5_/TiO_2_)_C200_ before and after the VL photocatalytic reaction appeared as strong light absorption bands at 578 nm and 584 nm, respectively. This was because the cationic Au and Cu species were reduced to a metallic Au and Au-Cu alloy, which exhibited strong absorption for the VL. The fresh (Au_1_Cu_0.5_/TiO_2_)_OP_ only exhibited a weak absorption band at 576 nm; however, the used sample exhibited a strong absorption band at around 602 nm after the VL PCO of CO due to an auto-reduction in the sample during the induction period [9]. Considering the strong interaction between Au and Cu species in the fresh (Au_1_Cu_0.5_/TiO_2_)_OP_, this auto-reduction process could generate a large amount of the Au-Cu alloy and, thus, make the used sample show higher absorption for VL via the strong LSPR effect. Figure 7a also showed that the absorption bands of the used bimetallic nanocatalysts shifted to a higher wavelength compared to the fresh samples, which was due to the formation of AuCu alloy nanoparticles during the VL PCO of CO. The absorption band of the used (Au_1_Cu_0.5_/TiO_2_)_OP_ centered at a higher wavelength than that of (Au_1_Cu_0.5_/TiO_2_)_C200_, indicating that O_2_ plasma treatment favored the formation of AuCu alloy nanoparticles. Further, we noted that the used (Au_1_Cu_0.5_/TiO_2_)_OP_ exhibited the broadest absorption band among all the samples, implying its fast rate in LSPR relaxation [41,42]. This facilitated the formation of hot electrons via Landau damping and the following direct injection of hot electrons to the TiO_2_ support [1,3], which could lead to an enhanced hot electron transfer process during the photocatalytic reaction. The photocurrent densities for different plasmonic nanocatalysts were also confirmed by the enhanced hot electron transfer on the (Au_1_Cu_0.5_/TiO_2_)_OP_. Figure 7b shows that all plasmonic nanocatalysts exhibited low current densities in the darkness but exhibited distinct photocurrent densities under VL irradiation. Compared to the monometallic samples, the higher photocurrent densities for bimetallic nanocatalysts suggested the favorable hot electron transfer on AuCu-TiO_2_ interfacial sites. As shown in Figure 7b, the (Au_1_Cu_0.5_/TiO_2_)_OP_ obtained the highest photocurrent density among all the plasmonic samples, which could be attributed to its high dispersion in the AuCu alloy nanoparticles (see the HR-TEM results) when considering the construction of large numbers of AuCu-TiO_2_ interfacial sites for hot electron transfer. Moreover, compared with the Au-TiO_2_ interface, the formation of the AuCu-TiO_2_ interface in (Au_1_Cu_0.5_/TiO_2_)_OP_ lowered the Schottky barrier height [43,44], which enabled more hot electrons to overcome the Schottky barrier under VL irradiation. Additionally, photocurrent densities for the plasmonic nanocatalysts seemed to show a gradual decrease with an increase in the test time (Figure 7b). To confirm this point, we performed a potentiostatic chronoamperometry study of O_2_ plasma-treated samples after the VL PCO of CO. All samples presented a decrease in their photocurrent density under VL irradiation during 30 min of continuous testing (see Appendix A). Compared with the (Au_1_/TiO_2_)_OP_ and (Cu_1_/TiO_2_)_OP_, (Au_1_Cu_0.5_/TiO_2_)_OP_ exhibited a steeper decrease in its photocurrent density, which could probably be attributed to the remarkable changes in its surface properties in the Na_2_SO_4_ solution when used for a potentiostatic chronoamperometry test.

## 3. Discussion

### 3.1. AuCu-TiO_2_ Interfacial Sites Construction

In this work, we demonstrate that the plasmonic Au-Cu/TiO_2_ nanocatalyst treated by O_2_ plasma showed a remarkable boost of photocatalytic activity in the VL PCO of CO compared with the counterparts. We also confirmed that the construction of AuCu-TiO_2_ interfacial sites during O_2_ plasma treatment played an important role in boosting the photocatalytic activity of Au-Cu/TiO_2_ nanocatalysts. The cationic Au and Cu species were randomly distributed on the surface of the TiO_2_ support for as-prepared Au-Cu/TiO_2_ bimetallic nanocatalysts (see Figure 8). During treatment, these cationic Au and Cu species occurred with redispersion and reduction simultaneously, which is essential for the formation of AuCu alloy nanoparticles and, thus, AuCu-TiO_2_ interfacial sites. It was noted that alteration behaviors of Au and Cu species in the treatment highly depended on the treatment approaches. Calcination could reduce most cationic Au species but hardly formed intimate interactions with the Au and Cu species because of their several migration and aggregations at high temperatures. By contrast, O_2_ plasma treatment enabled a strong interaction between the cationic Au and Cu species to form particles of mixed oxide AuCuO_x_, as O_2_ plasma established a unique redox environment on the nanocatalyst surface through large numbers of charged species [38]. The intimately interacted Au and Cu species benefitted from forming an AuCu alloy and, thus, constructing AuCu-TiO_2_ interfacial sites and also contributing to and controlling the size of AuCu alloy nanoparticles. These intimately interacted Au and Cu species in (Au_1_Cu_0.5_/TiO_2_)_OP_ could be auto-reduced into the AuCu alloy in the VL PCO of CO since the hot electrons produced by metallic Au nanoparticles and generated from O_2_ plasma treatment could quickly and efficiently reduce adjacent cationic Cu to form an AuCu alloy under VL irradiation (see Figure 8). This auto-reduction process was completed at the beginning of a photocatalytic reaction (within the induction period) [9]. Apparently, compared with O_2_ plasma, conventional calcination tended to separate the Au and Cu species, which suppressed the formation of AuCu alloy nanoparticles and, thus, the construction of AuCu-TiO_2_ interfacial sites.

### 3.2. The Boost of VL Photocatalysis on AuCu-TiO_2_ Interfacial Sites

The processes of VL PCO of CO over the Au-Cu/TiO_2_ bimetallic nanocatalyst could be described by a charge (hot electron) injection mechanism [3]. The AuCu alloy nanoparticles adsorbed CO molecules and also acted as a dye sensitizer to produce hot electrons by absorbing resonant photons under VL illumination. The energetic hot electrons were transferred to the TiO_2_ support via surmounting the AuCu-TiO_2_ interfaces. In addition, O_2_ molecules were adsorbed on the TiO_2_ support and combined with injected hot electrons to form superoxide (O_2_^−^). Finally, the O_2_^−^ dissociates reacted with the adsorbed CO molecules at AuCu-TiO_2_ interfacial sites to form CO_2_. It was demonstrated that the (Au_1_Cu_0.5_/TiO_2_)_OP_ possessing large numbers of coordinatively unsaturated sites not only facilitated CO adsorption (see CO chemisorption results) but also enhanced the formation of O_2_^−^ by generating abundant surface oxygen and hot electrons (see XPS and optical property results) due to the construction of large numbers of AuCu-TiO_2_ interfacial sites. This provides evidence that (Au_1_Cu_0.5_/TiO_2_)_OP_ exhibits advantages over the counterparts in providing reactants for VL PCO of CO.

For the processes of O_2_^−^ the dissociation and the following interfacial reactions between CO and O, we further discuss this based on DFT calculation results (see Appendix A). A unique feature of (Au_1_Cu_0.5_/TiO_2_)_OP_ is that it constructs large numbers of AuCu-TiO_2_ interfacial sites. On this plasmonic Au-Cu/TiO_2_ nanocatalyst, the dissociation barrier for O_2_^−^ showed a much higher value (0.64 eV) than the barrier (0.15 eV) for CO to react with dissociated O at the AuCu-TiO_2_ interfacial sites (see Figure 9). These two elementary reaction processes arrived at similar conclusions on the plasmonic Au/TiO_2_ and CuOx/TiO_2_ nanocatalysts (see Figure 9), indicating the rate-limiting step of O_2_^−^ dissociation. The (Cu_1_/TiO_2_)_OP_ presented very poor VL photocatalytic activity, as it had a high dissociation barrier for O_2_^−^ (0.78 eV) and weak LSPR to generate O_2_^−^ (see the UV-vis DRS results). Interestingly, the DFT results for (Au_1_Cu_0.5_/TiO_2_)_OP_ and (Au_1_/TiO_2_)_OP_ indicate that they possess a comparable barrier for O_2_^−^ dissociation (see Figure 9a), implying that the construction of AuCu-TiO_2_ interfacial sites showed a weak influence on the O_2_^−^ dissociation process. By contrast, the construction of AuCu-TiO_2_ interfacial sites remarkably reduced the barrier for a reaction between CO and O from 0.42 eV to 0.15 eV in comparison with (Au_1_/TiO_2_)_OP_ (see Figure 9b). The above analyses, taken together, suggest that the boost of photocatalysis of the (Au_1_Cu_0.5_/TiO_2_)_OP_ in VL PCO of CO originated from the strong interaction between the Au and Cu species derived from O_2_ plasma treatment. This strong interaction controlled the size of AuCu alloy nanoparticles, establishing larger numbers of interfacial sites for the hot electron transfer and CO oxidation reaction. Moreover, the intimately interacted Au and Cu species favored the formation of an AuCu alloy, which boosted the hot electron transfer process by lowering the Schottky barrier height and also significantly reduced the barrier for CO to react with O.

## 4. Materials and Methods

### 4.1. Catalysts Preparation

The plasmonic Au-Cu/TiO_2_ bimetallic nanocatalysts were prepared using TiO_2_ (P25, Degussa) as support and HAuCl_4_ and Cu(NO_3_)_2_ as precursors of plasmonic metals [45]. The solutions of HAuCl_4_ (2.43 × 10^−2^ mol/L; 2.2 mL) and Cu(NO_3_)_2_ (4.14 × 10^−2^ mol/L; 3.8 mL) were mixed with 1.0 g of TiO_2_, and the mixture was stirred with a glass rod at room temperature for 10 min. Subsequently, the samples were aged for 14 h at room temperature and then washed with deionized water and an aqueous ammonia solution (pH 8). After washing, the samples were dried at 80 °C for 6 h to obtain the as-prepared Au-Cu/TiO_2_ nanocatalyst. Unless otherwise specified, the nominal loadings of Au and Cu in as-prepared Au-Cu/TiO_2_ nanocatalysts were 1 wt.% and 0.5 wt.%, respectively. For comparison, TiO_2_-supported Au (nominal 1 wt.%) and Cu (nominal 1 wt.%) nanocatalysts were prepared using the same procedures.

The as-prepared plasmonic nanocatalysts (about 30 mg) were coated on a glass substrate of 50 mm (L) × 25 mm (W) × 1 mm (T) [13] and were treated by calcination (200 °C in the air for 2 h) or cold plasma before the photocatalytic activity tests. The plasma treatment was performed in a homemade dielectric barrier discharge (DBD) reactor [9,15]. Our previous work demonstrated that O_2_ plasma outperformed other plasmas in plasmonic nanocatalyst activation [13], and thus, all plasma treatment (AC discharge; 1.8 kHz; CTP-2000 K, Nanjing Suman Electronics Co., Nanjing, China) experiments were carried out for 10 min at an input power of 10 W with an O_2_ flow rate of 100 mL/min. The Au_1_Cu_0.5_/TiO_2_ treated by calcination and O_2_ plasma were denoted as (Au_1_Cu_0.5_/TiO_2_)_C200_ and (Au_1_Cu_0.5_/TiO_2_)_OP_, respectively. The Au_1_/TiO_2_ and Cu_1_/TiO_2_ treated by O_2_ plasma were denoted as (Au_1_/TiO_2_)_OP_ and (Cu_1_/TiO_2_)_OP_, respectively.

### 4.2. Catalysts Characterization

The actual loadings of Au and Cu in the nanocatalysts were determined by inductively coupled plasma-atomic emission spectroscopy (ICP-AES, Optima 2000DV, PerkinElmer Inc., Waltham, MA, USA). The Au loading for Au_1_/TiO_2_ was 0.84 wt.%, and the Cu loading for the Cu_1_/TiO_2_ was 0.78 wt.%. The Au and Cu loadings for Au_1_Cu_0.5_/TiO_2_ were 0.82 wt.% and 0.43 wt.%, respectively. Temperature-programmed reduction with H_2_ (H_2_-TPR) was performed in an AutoChem 2910 system (Micromeritics, Atlanta, GA, USA). The fresh samples (100 mg) were placed in a quartz reactor and pretreated in helium (50 mL/min) for 30 min at −50 °C. Then, 5% H_2_ in helium (50 mL/min) was switched to flow through the pretreated samples at −50 °C until there was no change in the signals. After that, the samples were heated to 250 °C at a rate of 5 °C min^−1^. The X-ray photoelectron spectroscopy (XPS) of the samples was performed at ESCALAB250 (Thermo VG, Waltham, MA, USA) with operating conditions of 15 kV and 300 W. The binding energies were calibrated with reference to the peak of C 1s at 284.6 eV. The in situ diffuse reflectance infrared Fourier transform spectra (DRIFTS) of CO adsorption were recorded by an FT-IR spectrometer (Nicolet 6700, Thermo Fisher, Waltham, MA, USA) at a resolution of 4 cm^−1^. Before the measurement, the samples placed in the DRIFTS cell were pretreated with N_2_ (100 mL/min) for 30 min at 80 °C. Subsequently, the samples were cooled down to room temperature and collected with background spectra in N_2_ flowing (100 mL/min). Finally, CO adsorption tests were performed by switching 1% CO/N_2_ (50 mL/min) into the DRIFTS cell. The plasmonic nanoparticles of the samples were observed by transmission electron microscopy (TEM; Tecnai G2F30 STWIN, Hillsboro, OR, USA) at 300 kV equipped with a high-angle annular dark-field (HAADF) detector and energy dispersive X-ray spectrometer (EDXS). The UV-vis diffuse reflectance spectroscopy (UV-vis DRS) of the samples was performed at a lambda 750s spectrometer (Perkin-Elmer, Waltham, MA, USA) to measure their photo-response properties. The photocurrent of the samples was measured using a three-electrode cell (0.1 M Na_2_SO_4_ solution) equipped with an electrochemical workstation (CHI660E, Chenhua, Shanghai, China). An ITO (indium tin oxide) glass of 20 mm (L) × 20 mm (W) × 1 mm (T) coated with the samples was used as a working electrode, while a platinum sheet and an Ag/AgCl electrode were used as the counter and reference electrodes, respectively. The photocurrent of the samples was measured at a bias voltage of 0.5 V under VL irradiation (58 mW/cm^2^).

### 4.3. DFT Calculation

The VL PCO of CO was investigated by Vienna Ab initio Simulation Package (VASP) calculations, in which the Au/TiO_2_, Au-Cu/TiO_2,_ and CuO_x_/TiO_2_ were selected as the calculation models for the comparison. The details of the computational method are described in the Appendix A.

### 4.4. Photocatalytic Reaction Evaluation

To evaluate the performance of the plasmonic nanocatalysts, the VL PCO of CO was performed in a home-made plate reactor [9,13], and VL with a light intensity of 58 mW/cm^2^ was supplied by a LED lamp [13]. The reactant gas composed of synthetic air (80% N_2_ + 20% O_2_) and 920 ppm CO was introduced into the plate reactor at a total flow rate of 300 mL/min. The concentrations of CO_x_ (CO and CO_2_) were monitored online by a CO_x_ analyzer (GXH-3011N, Huayun, Beijing, China), and CO conversion (*X*_CO_) was defined as Equation (1):(1)$$X_{\mathrm{CO}}=\frac{C_{{CO}_{2}}}{C_{\mathrm{CO}}}$$
where $C_{\mathrm{CO}}$ and $C_{{CO}_{2}}$ denote the concentrations of CO in the inlet gas and CO_2_ in the outlet gas, respectively.

## 5. Conclusions

In conclusion, this work demonstrates that the plasmonic Au-Cu/TiO_2_ bimetallic nanocatalyst, prepared with O_2_ plasma treatment, could exhibit superior photocatalytic activity under VL irradiation. The O_2_ plasma treatment boosted the VL photocatalytic activity of the Au-Cu/TiO_2_ nanocatalyst by nearly 50% compared to the conventional calcination treatment. After O_2_ plasma treatment, the Au-Cu/TiO_2_ nanocatalyst showed comparable performance to the monometallic Au/TiO_2_ sample possessing double Au loading. The O_2_ plasma treatment could initiate the surface alteration of the Au-Cu/TiO_2_ nanocatalyst to generate a strong interaction between the Au and Cu species. The intimately interacted Au and Cu species decreased their reduction temperature and, thus, were quickly auto-reduced into AuCu alloy nanoparticles at the beginning of the VL PCO of CO. The average size of plasmonic nanoparticles for the O_2_ plasma-treated Au-Cu/TiO_2_ nanocatalyst was controlled at around 2.4 nm by the interaction between Au and Cu species, creating larger numbers of interfacial sites for CO adsorption and oxidation in comparison with the conventional calcined sample. Compared with the monometallic plasmonic nanocatalyst, the formation of small AuCu alloy nanoparticles reduced the Schottky barrier height at the metal–support interface and constructed highly active AuCu-TiO_2_ interfacial sites, which facilitated the interfacial transfer of hot electrons. The DFT calculations confirmed that the construction of AuCu-TiO_2_ interfacial sites contributed to a low barrier for the reaction between CO and O and, thus, the boosted VL photocatalytic activity, although O_2_^−^ dissociation was the rate-limiting step for the VL PCO of CO.

## Figures and Tables

**Figure 1 ijms-24-10487-f001:**
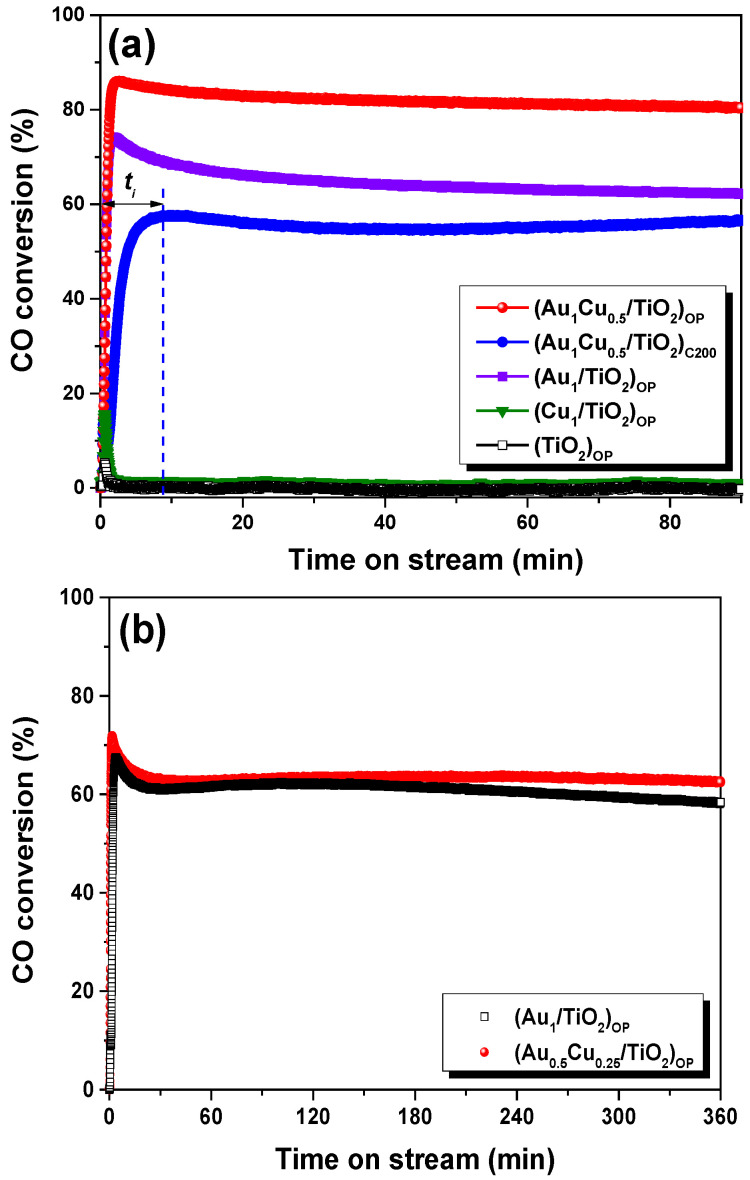
(**a**) CO conversion as a function of time under VL irradiation over various plasmonic nanocatalysts, and (**b**) a comparison of VL photocatalytic activity for the (Au_1_/TiO_2_)_OP_ and (Au_0.5_Cu_0.25_/TiO_2_)_OP_. *t_i_* represents the induction period.

**Figure 2 ijms-24-10487-f002:**
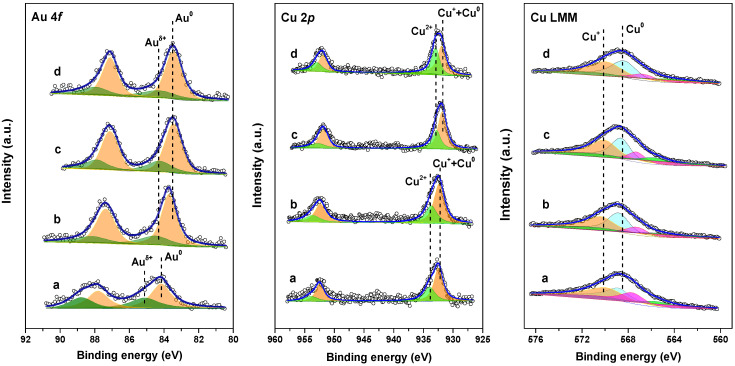
XPS Au 4*f*, and Cu 2*p* and Cu LMM of the fresh (**a**) and used (**b**) (Au_1_Cu_0.5_/TiO_2_)_OP_, and the fresh (**c**) and used (**d**) (Au_1_Cu_0.5_/TiO_2_)_C200_.

**Figure 3 ijms-24-10487-f003:**
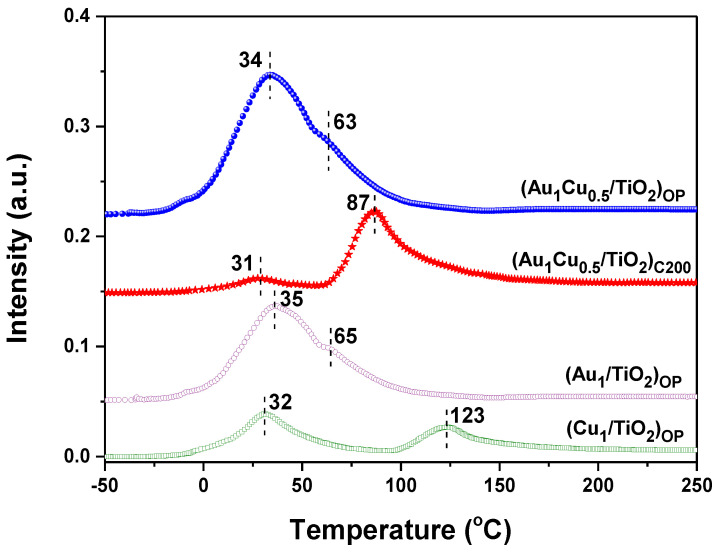
H_2_-TPR profiles of fresh plasmonic nanocatalysts.

**Figure 4 ijms-24-10487-f004:**
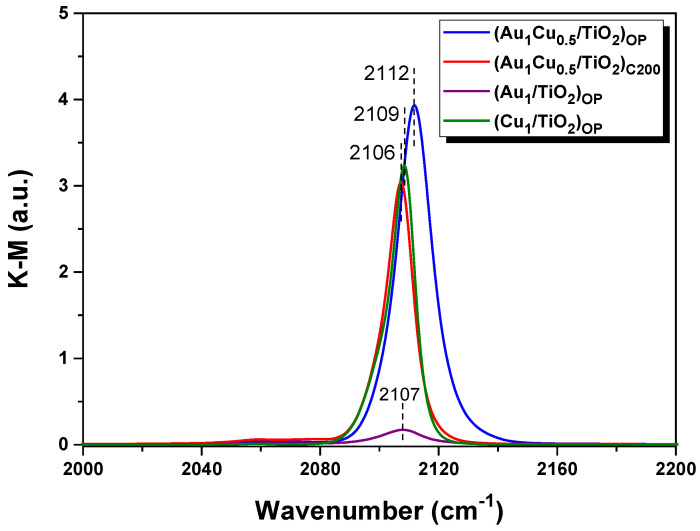
In situ DRIFTS of CO adsorption over various plasmonic nanocatalysts at an exposure time of 30 min.

**Figure 5 ijms-24-10487-f005:**
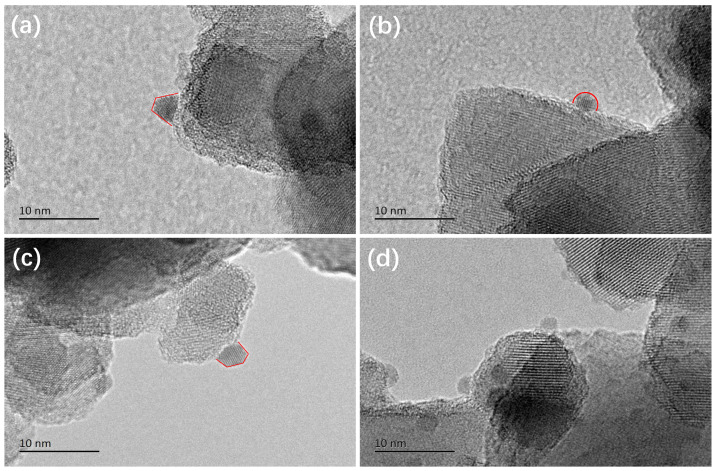
HR-TEM images of the used (**a**) (Au_1_Cu_0.5_/TiO_2_)_OP_, (**b**) (Au_1_Cu_0.5_/TiO_2_)_C200_, (**c**) (Au_1_/TiO_2_)_OP_, and (**d**) (Cu_1_/TiO_2_)_OP_. The red lines highlight the shape of metallic nanoparticles. The used nanocatalysts were obtained by applying fresh samples to the VL PCO of CO for 90 min.

**Figure 6 ijms-24-10487-f006:**
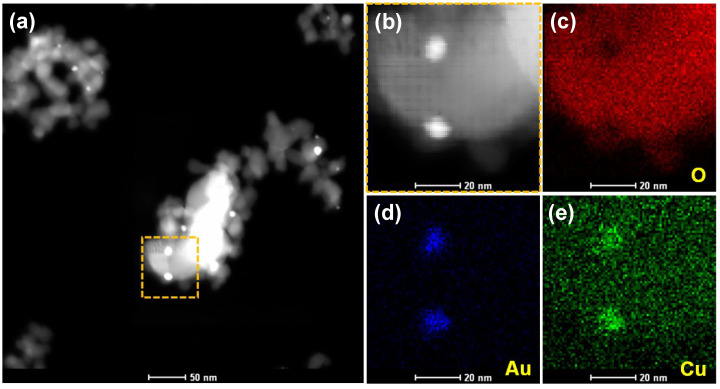
(**a**) General HAADF-STEM image of the used (Au_1_Cu_0.5_/TiO_2_)_OP_, and (**b**) enlarged HAADF-STEM image and (**c**–**e**) the corresponding element mapping patterns of the bimetallic nanoparticle. The dotted frames highlight the location of the enlarged image in a. The used nanocatalyst was obtained by applying the fresh sample to the VL PCO of CO for 90 min.

**Figure 7 ijms-24-10487-f007:**
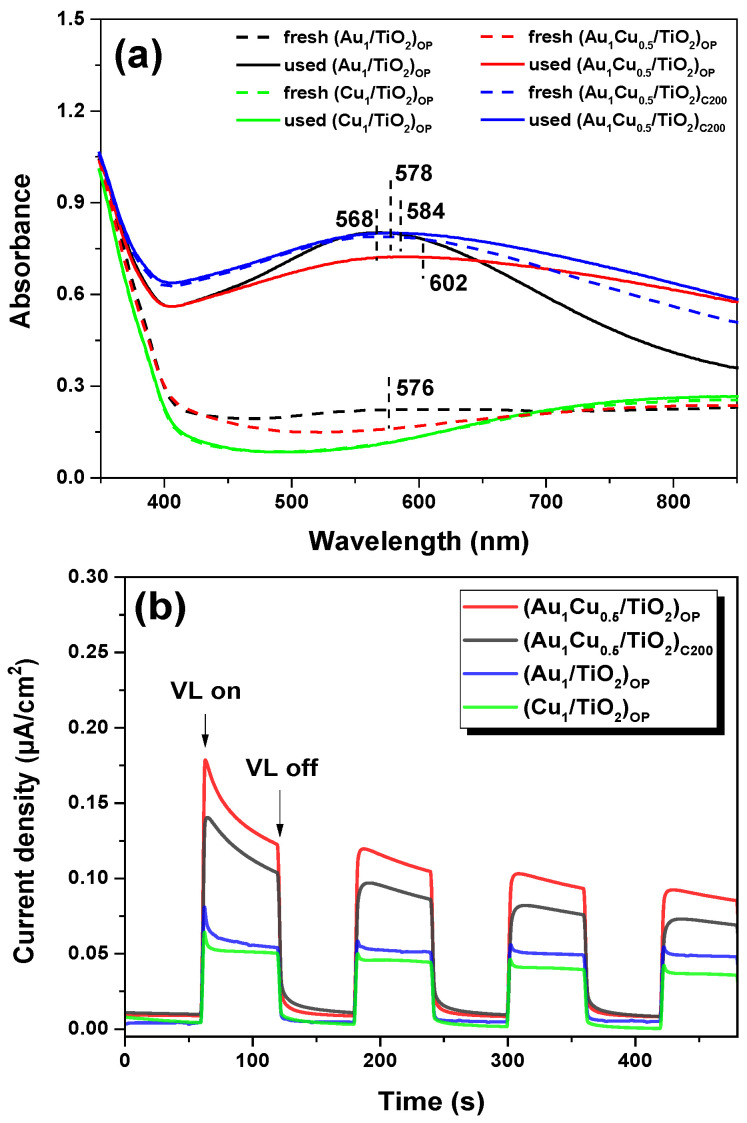
(**a**) UV-vis DRS of fresh and used plasmonic nanocatalysts, and (**b**) Photocurrent density for used plasmonic nanocatalysts under VL irradiation. The used nanocatalysts were obtained by applying the fresh samples to the VL PCO of CO for 90 min.

**Figure 8 ijms-24-10487-f008:**
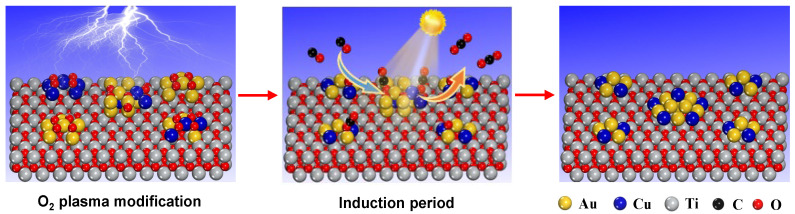
Schematic diagram for the construction processes of AuCu-TiO_2_ interfacial sites by O_2_ plasma treatment for the plasmonic Au-Cu/TiO_2_ bimetallic nanocatalyst.

**Figure 9 ijms-24-10487-f009:**
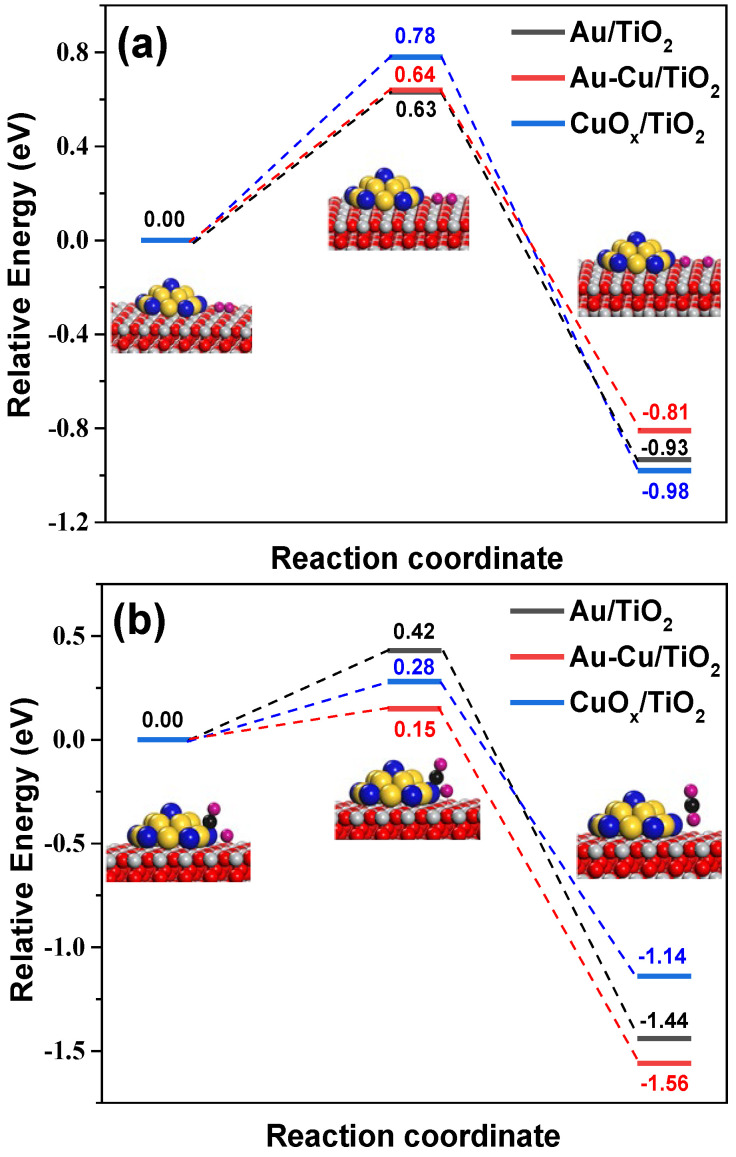
Calculated activation barriers for (**a**) The dissociation of O_2_^−^ and (**b**) CO to react with O on different plasmonic nanocatalysts. Inset: O_2_^−^ dissociation and reaction between CO and O on Au-Cu/TiO_2_ model interfaces. Color code: Au, golden; Ti, grey; O atom in TiO_2_, red; O atom in O_2_, CO and CO_2_, pink; Cu, blue; C, black.

**Table 1 ijms-24-10487-t001:** XPS analysis of the plasmonic bimetallic samples.

Sample	Au 4*f*_7/2_ (eV)		Cu 2*p*_3/2_ (eV)		Cu LMM (eV)		Au^δ+^/Au (at.%)	Cu^2+^/Cu (at.%)	Cu^0^/Cu^+^
	Au^δ+^	Au^0^	Cu^2+^	Cu^0^ + Cu^+^	Cu^0^	Cu^+^			
a	85.1	84.2	934.0	932.5	568.7	570.1	43	35	0.88
b	84.5	83.7	933.8	932.4	568.8	570.2	29	32	0.97
c	84.2	83.4	932.8	931.9	568.7	570.3	38	40	0.83
d	84.3	83.4	933.0	932.0	568.5	570.0	24	36	0.91

Note: Au = Au^0^ + Au^δ+^; Cu = Cu^0^ + Cu^+^ + Cu^2+^; a and b denote the fresh and used (Au_1_Cu_0.5_/TiO_2_)_OP_, respectively; c and d represent the fresh and used (Au_1_Cu_0.5_/TiO_2_)_C200_.

**Table 2 ijms-24-10487-t002:** The number of perimeter, total atoms and the proportion of perimeter atoms for the (Au_1_Cu_0.5_/TiO_2_)_OP_, (Au_1_Cu_0.5_/TiO_2_)_C200_ and (Au_1_/TiO_2_)_OP_.

Samples	$N_{p}$	$N_{t}$	Proportion of Perimeter Atoms (%)
(Au_1_Cu_0.5_/TiO_2_)_OP_	16	267	6.0
(Au_1_Cu_0.5_/TiO_2_)_C200_	28	601	4.7
(Au_1_/TiO_2_)_OP_	30	564	5.3

Note: *N*_p_ and *N*_t_ are the numbers of the perimeter and total atoms, respectively.

## Data Availability

The data presented in this study are available upon request from the corresponding author.

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
