# Peer review of "Boosting the Photocatalysis of Plasmonic Au-Cu Nanocatalyst by AuCu-TiO2 Interface Derived from O2 Plasma Treatment"

_ijms, 2023, doi:10.3390/ijms241310487_

Round 1

Reviewer 1 Report

The authors show an O2 plasma treatment of Au based catalysts to improve their photocatalytic activity for CO conversion. The work needs some further work before being considered for publication.

1.       English writing needs to be checked. Please correct expressions such as “The workers”…

2.       Introduction should mention why photocatalysis is a viable solution for CO conversion.

3.       Introduction should mention why CO conversion was investigated.

4.       If the structure of the article is to remain like this, the catalysts nomenclature must be given in the beginning of the results, so a reader can follow the text.

5.       How were HAuCl4 and Cu(NO3)2 impregnated into TiO2?

6.       How were the 1%Au and 0.5%Cu selected? Is this the optimal ratio?

7.       What is the reasoning behind the peak of CO conversion with the Cu1/TiO2 catalyst?

8.       Control experiments with TiO2, with no VL and with the metals alone should be given.

9.       Please merge Figs 1a and 1b. Add bare TiO2.

10.   Why not investigate higher loading of Au, for example Au2Cu0.5/TiO2-OP, or other ratios such as Au1.5Cu1, Au1Cu0.25…?

11.   Why not treat Au/TiO2 and Cu/TiO2 be C200 treatment?

12.   In XPS, it it necessary to better highlight and compare the monometallic OP catalysts and the Au1Cu0.5 material. Are there significant shifts, they will show the interaction.

13.   How do you explain the lack of Cu species peak in the TPR of the AuCu catalyst?

14.   Transient photocurrent seems to show loss of activity. Please provide a potentiostatic chronoamperometry study of used and fresh catalysts to show the stability.

15.   Provide a stronger argument for higher absorption after CO conversion.

Some English changes should be made to ease reading. 

Reviewer 2 Report

Comments on the manuscript “Boosting the photocatalysis of plasmonic Au-Cu nanocatalyst by AuCu-TiO2 interface derived from O2 plasma treatment” by B. Zhu, X. Li, Y. Li, J. Liu, X. Zhang, submitted to MDPI International Journal of Molecular Sciences.

The authors describe an attempt to improve the performance of the Au-based plasmonic catalysts, first by introducing a Cu admixture to the Au-nanoparticles, and secondly, by applying an innovative pretreatment of the catalysts based on using the cold O2 plasma, rather than more conventional calcination step at 200 °C.  The success of this strategy is then demonstrated in the photocatalytic oxidation (PCO) of CO.  The authors also discuss a possible mechanism of the effectiveness of the O2 plasma in action.  I find the paper publishable, but after the authors have given thought to the following points.

1) The proposed mechanism of O2 plasma action.

I find it highly unusual to suggest, that the oxygen plasma may create a reducing environment where it is applied.  But that is what we read in the Discussion, pages 302 – 310.  To suggest that O2 plasma reduces cationic forms of Au and Cu seems to go against not only a simple chemical intuition, but also against basic chemical knowledge as well, since the atomic oxygen is considered to be a strong oxidizing agent, by no means a reducing one.  Personally, I think that the O2 plasma treatment actually oxidizes the surface Au and Cu species, and doing so, it leads to the formation of particles of a mixed oxide AuCuOx.  Such mixed oxide particles may then undergo reduction in situ during reaction, forming the well alloyed AuCu nanoparticles that are crucial for the catalytic performance of this material.

2) The HR-TEM images.

On page 6, the authors state: “The HR-TEM images confirm that the plasmonic nanocatalysts treated by O2 plasma can obtain metallic plasmonic nanoparticles with small particle size and thus the coordinatively unsaturated sites (such as edges and corners).” I am afraid, I must disagree. In my opinion, the only thing that can be safely concluded from the images presented is that there are certain nanoparticles on the TiO2 surface and, moreover, they all are of roughly the same size.  And this seems to be a nontrivial piece of information on its own merit, to be obtain for those images.  The discussion it terms of whether the small nanoparticles are displaying or not some corners and edges is an overstretch, given the presented data.

3) Some minor points

- Fig 7a, the legend.  The green dashed line is labeled as representing used (Cu/TiO2), but it seems it should be the fresh (Cu/TiO2)

- Fig 2. Would the authors please enlarge the figures at least a little bit.

The English of the paper is generally good and comprehensible, but there seem to be some cases of strange usage, for instance (line 300): “The cationic Au and Cu species randomly distribute on the surface of TiO2 support [...]”; here I would suggest using the verb in passive voice “are randomly distributed”.  Another example (line 166): “the (Au1Cu0.5/TiO2)C200 appears the reduction peak at” should rather read “the (Au1Cu0.5/TiO2)C200 shows (or exhibits) the reduction peak at”.  Yet another one (line 437): “bimetallic 437 nanocatalyst, characterized by O2 plasma treatment,” should rather be “bimetallic nanocatalyst, prepared with O2 plasma treatment,”.

Round 2

Reviewer 1 Report

The authors failed to provide details about other quantities of Au, Cu or TiO2 or different ratios in the different treatments, focusing on only 1% Au based of the literature. This should be corrected in future publications when comparing catalysts with several variable parameters, that have to be clearly explained and optimised. However, they answered to all of my concerns regarding the review report, and therefore the manuscript can be considered for publication.